# Approach to an Equivalent Freight-Based Sustainable Joint-Quotation Strategy for Shipping Blockchain Alliance

**Fa Zhang and Yimiao Gu ***

Department of Electronic Business, South China University of Technology, Guangzhou 510006, China
* Correspondence: guymcheers@scut.edu.cn

**Abstract:** To improve the sustainability of the shipping industry, a practice of establishing a new type of shipping alliance based on blockchain has been implemented. In this practice, the following question emerges: How will shipping lines achieve sustainable profit improvement? This paper focuses on the freight decision-making problem by constructing a multi-round joint-quotation strategy. This paper also demonstrates the potential impact of a joint quotation strategy on the blockchain-based open freight market from a theoretical perspective. The numerical experiment results show that, compared with the initial state, the joint quotation strategy can help to stabilize shipping demands and reduce the fluctuation in overall demands. In this strategy, the freight level needs to be high to maximize profits. However, part of the demands will be sacrificed as a result. Moreover, the optimal equilibrium solution under the joint quotation strategy is relatively vulnerable to changes in the competitive relationship among the members of an alliance. In addition, the joint quotation may also be resolutely resisted by the shipper due to monopoly risk, resulting in a major risk of a sharp reduction in demand. The findings in this paper offer a decision-making reference for the sustainable development of the shipping industry.

**Keywords:** shipping blockchain alliance; joint quotation; container shipping line; equivalent freight; sustainability





## 1. Introduction

In recent years, the development of the shipping industry has shown a general fluctuating trend. Since the global economic crisis in 2008, the shipping industry has experienced a long difficult period due to a continuous downturn in demand [1,2]. For a time, shipping lines had to lower the freight rate because of excess capacity. Additionally, at the same time, the quality of service was unable to meet the demand of the shippers [3]. However, the shipping industry, especially the container-shipping industry, has rebounded rapidly since 2020. In a very short period of time, the previous phenomenon of excess capacity has changed, such that there is presently a short supply of container-shipping services.

Neither a serious excess in transport capacity [4] nor an excessive supply of shipping services is conducive to the long-term sustainable development of the shipping industry [5]. These extreme situations may cause the shipping lines to run into difficulties in shipping service pricing. As the global trade continues to be depressed, and the service supply of the shipping industry seriously exceeds the container shipping demand, shipping lines have to reduce freight in order to maintain their operation, according to general economic principles. At the time, when the freight rate is too low [6], there will be a direct negative impact on competitors, eventually leading to vicious competition in the original oligopoly market. Although there is no possibility of vicious competition at a low freight rate in cases of short supply, high freight rates will also discourage shippers [7]. As the demand is too high, the limited transport capacity cannot ensure that the cargoes can be delivered to the destination in time. Shippers may prefer to choose other alternative transportation methods in the face of high freight rates. Therefore, reasonable pricing and stable demand are conducive to the long-term sustainable development of the container-shipping industry.

On the one hand, stable demand requires reasonable pricing by shipping lines. Shipping line operations should not only maximize their profits, but also be conducive to the long-term sustainable development of the industry. Therefore, how should a long-term freight rate strategy that is conducive to stable and sustainable development be built? First of all, shipping lines need to identify the main factors affecting the long-term development of the industry. These factors include: (1) the market risk faced by the container-shipping industry; and (2) the pricing decision failure and lag caused by large demand fluctuations [4]. Additionally, they can improve their risk–response ability through alliances. Alliance strategies have a significant impact on the container-shipping industry [8]. Furthermore, cooperation among alliance members should be strengthened [9], and a community of common destiny needs to be built. Furthermore, more reasonable freight strategies need to be adopted. Shipping lines need to solve a series of problems, such as the limitation of profit potential caused by low freight, and the difficulty of guaranteeing service quality, which is also accompanied by low freight [10].

On the other hand, the industry must improve the efficiency of container-shipping services in the interest of guaranteeing stable demand. Practice has proved that blockchain plays an increasingly important role in the transformation of business processes in the shipping industry [11–13]. Decentralized consensus mechanisms, smart contracts [11], and confidential information transmission mechanisms can help the container-shipping industry to achieve the efficient paperless management of documents. In particular, the distributed ledger function can ensure the permanent preservation and traceability of reliable logistical information. These functions are greatly improving the operational efficiency of the container-shipping industry. In addition, theoretical research shows that the construction of shipping blockchain public-quotation mechanisms can improve the current situation of shipping lines. Moreover, it may help shipping lines to obtain increased profits compared with the traditional market and stabilize demand at the same time.

Although blockchain is beneficial for container-shipping members, the following question arises: how will members make decisions from the perspective of stabilizing alliance demand and maximizing alliance profit? One consideration is that container-shipping blockchain alliance members can strengthen cooperation and adopting the joint quotation strategy [14]. Here, the joint quotation strategy for members of a container-shipping blockchain can aid in assuring contracts. Additionally, such an approach can formulate a unified standard for freight decision making with the goal of maximizing alliance profit and realizing their own increased profits.

With the maturation of blockchain technology and the accelerating integration of container-shipping service supply chains, the disclosure of freight information will become a development trend. In such a situation, the following questions arise: Will the further strengthening of cooperation be beneficial for shipping lines in achieving their goals of stabilizing demands and improving profits? Is this strengthened cooperation feasible in reality and conducive to the long-term sustainable development of the shipping blockchain alliance? Based on the popular shipping blockchain alliance, this paper presents a feasible quotation strategy for alliance members on the basis of no freight information disclosure in an emerging container-shipping blockchain. Then, the feasibility of establishing freight alliances from the perspectives of profit and demand is discussed. The paper is organized as follows: Section 2 presents a literature review on market risk, shipping alliances, joint quotations, and shipping blockchains. Section 3 discusses the binary strategy, based on a multi-round freight quotation approach. Section 4 consists of numerical experiments and sensitivity analyses. Section 5 concludes the research.

## 2. Literature Review

The literature about three aspects is reviewed in this part, including the shipping alliance, shipping blockchain, and joint quotation.

The existing literature shows that scholars' attention to the shipping industry has gradually focused on the shipping alliance. Some scholars have demonstrated in the way of theoretical research that the alliance strategy will have a positive impact on the development of container shipping industry. Chen et al. [15] found that there are research opportunities in the diversification and flexibility of cooperation modes and intelligent operation in liner shipping alliance management through a systematic review of liner shipping alliance management. Do et al. [16] confirmed that when the internal competition in the shipping industry intensifies, reducing the number of new ship orders in time and taking alliance strategic actions at the same time will help to rebalance the shipping industry. Chiu and Wang [17] proved that the utilization rate of production factors will be improved when liner shipping lines provide services through a strategic alliance. However, present research on shipping alliance is mainly based on the perspective of vertical integration of shipping service supply chain [18–21]. Wang and Wang [22] focused on the vertical alliance decision-making of ports and carriers from the perspective of green supply chain. Tran [23] reviewed the structural transformation in the container shipping industry from 1995 to 2020 and found that the establishment of the alliance has become an effective means for container shipping lines to deal with the reasonable competition of powerful competitors. Ma et al. [24] investigated the impact of vertical integration between shipping lines and ports on alliance stability by using a non-cooperative merger control game. At present, scholars pay more attention to the vertical integration of shipping services based on the blockchain. There is a lack of research in the field of competition and cooperation between homogeneous shipping lines in the same alliance. This problem is becoming more and more prominent when the shipping industry is facing a volatile external environment.

Theoretical research shows that constructing an alliance on the blockchain can break the information barrier of the traditional shipping industry and help to improve the operational efficiency of the shipping alliance. Tijan et al. [25] made a literature review on the driving factors, success factors, and obstacles of the digital transformation of the shipping industry. Influenced by similar research with Tijan, there have been many scholars having begun to pay attention to the theoretical feasibility of the application of blockchain in the shipping industry [26–30]. Ahn et al. [31] verified the importance and urgency of intelligent technologies, such as blockchain, in the shipping industry. Additionally, more scholars have started to explore the practical application of blockchain technology in the shipping industry. Hvolby et al. [32] studied and confirmed that the efficiency of blockchain information exchange mechanism in simplifying shipping business processes is very high. Pu and Lam [33] studied the introducing time of the blockchain applying the game theory. Wang et al. [34] investigated the trust mechanism between business entities of building a large-scale shipping network on blockchain. Baygin et al. [35] studied and built blockchain solutions for local shipping industry from the perspective of technology coupling. Ahn, Kim, Kim and Lee [31] verified the effectiveness of intelligent technologies related to blockchain in improving the operational efficiency of the shipping industry. However, the present literature on the shipping blockchain alliance is very scarce. Additionally, the existing literature only focuses on the blockchain based cooperation between carriers in the same supply chain. Hu and Dong [36] investigated the cooperative decision-making problem of container carriers with upstream and downstream relationships based on blockchain. Nowadays, there are more and more researches on the application of blockchain functions to solve business problems in the shipping field. However, there are very few instances in the literature focusing on the future sustainable development strategy selection of the shipping blockchain alliance and the horizontal competitive relationship among the members of the shipping blockchain alliance.

Theoretically, the existing studies have proved the feasibility of joint quotation strategy for company alliance. Sousa and Bradley [37] showed that the strategy of joint quotation is a reasonable and necessary market behavior for shipping lines as long as it is accepted by shippers. The condition lies in the premise of improving service quality, not just for profit. According to Sousa's study, the increase in joint freight can affect the network performance of shipping lines. With the change in joint freight, there is no additional cost, so the efficiency can be improved. Sabri et al. [38] pointed out that the efficiency improvement brought by joint quotation can help service providers (such as shipping lines) to eliminate the negative reaction of demanders (such as shippers) to the service providers strengthening the alliance. Hani and Dagnino [39] showed that, compared with individuals without alliance foundation, it is much easier for companies with alliance foundation to achieve joint quotation. It is because they can obtain higher benefits with lower risks and implementation costs. Chiao, Lin and Huang [14] analyzed the world's top 21 shipping lines through the structured content analysis method and made an important discovery. This is the premise of joint quotation, which is that shipping lines must strengthen cooperation. Another joint bidding problem that scholars are concerned about is mainly based on the joint behavior between vertical enterprises with upstream and downstream cooperation in the supply chain [40–42]. Therefore, joint quotation is a feasible market behavior, which helps to improve the operating efficiency of shipping lines and improve the profit, but it can be realized only on the basis of a certain alliance, especially on the basis of supply chain. There is a lack of research on joint quotation behavior among members of the shipping blockchain alliance, especially container shipping lines with horizontal competition.

The theoretical research shows that joint quotation is an important way to deal with market risks and improve the sustainability and stability of the alliance, especially today when the shipping blockchain is gradually mature, the operation mechanism of the shipping alliance on the blockchain is gradually improved. Additionally, the cooperation among members of the shipping blockchain is gradually strengthened. This has provided a good foundation for researching on freight decision and profit distribution of shipping alliance. Therefore, building a joint quotation strategy for an emerging container shipping blockchain is the focus of this paper.

## 3. Model Development

The theoretical research has verified that the public quotation of shipping blockchain cannot only regulate the behavior of the container shipping market, but also help container shipping lines to improve their profits. Therefore, the shipping blockchain alliance members are quite willing to reach the consensus of public quotation. On the basis, this paper mainly focuses on whether SLs, as members of the shipping blockchain alliance, are willing to further strengthen cooperation. The SLs will try to maximize the total profit of the container shipping alliance based on blockchain by signing a joint quotation agreement. Additionally, the adjustment of the freights of the members is a joint action [14]. It is a spontaneous behavior dominated by larger container SLs without mandatory. As a result, every member will obtain the chance of default to win excess return and the default risk of other members. The method of equal proportional adjustment of freight is adopted in this paper. That means, freights of all SLs will change proportionally according to the agreement.

So, does the research scenario of shipping blockchain open market have practical significance? The answer is yes. An important premise of the problem studied in this paper is that the emerging public quotation market based on shipping blockchain is gradually maturing. It does not mean the traditional container shipping market having been replaced. Especially for SLs as shipping blockchain members, there will be two competition contexts, traditional market and blockchain market. Moreover, the public quotation of shipping blockchain market is a spontaneous behavior based on strategic optimization for container shipping lines, which is non-mandatory. There is something interesting for the shipping blockchain market. (1) Firstly, the information confidentiality mechanism of blockchain and its strong exclusivity to non-members make the shipping blockchain market look more

like an independent market isolated from the traditional shipping market. That is, the information shared on the shipping blockchain will not be obtained by non-member of blockchain. (2) Additionally, the decision making in the shipping blockchain market is not antagonistic or conflicting with the tradition market. The members of shipping blockchain can still make decisions in traditional market. (3) Different from the general information interaction platform, the physical basis of the information interaction mechanism of shipping blockchain is the Consortium Blockchain rather than Public Chain. That means members on shipping blockchain can reach a consensus on the necessary information disclosure and interaction. For the scope of information disclosure, each business entity is optional, and not all information will be submitted publicly. When a container shipping item is written into the blockchain by the SL, it will be decrypted and verified by the relevant members. After being verified, the service information is stamped with a time stamp, and the new block is formed and connected with the existing block to realize the permanent storage of the service information on the blockchain.

Additionally, why is the container shipping market the only focus of this paper? It is mainly because the container shipping market has a relatively more standardized pricing mode, the mode of Box Rate. The freight rate of bulk cargo is usually affected by the weight and volume of cargo and other characteristics and seasons. The container shipping freight rate is unified with the container as the billing unit. Therefore, it is easier for container shipping lines to form a direct competitive relationship in freight decision making in the blockchain open quotation market.

To simplify the problem, two SLs are introduced into blockchain container shipping market in this paper. They are two SLs including Shipping Lines A (SLA) and Shipping Lines B (SLB). SLA is made to be larger than SLB. That means SLA takes a stronger scale economy effect than SLB. Additionally, the unit cost $c_1$ of SLA is lower than unit cost $c_2$ of SLB. So, SLA dominates in the traditional market. Both SLs have reached a consensus on writing freights on the shipping blockchain. Additionally, the freight can be shared between SLA and SLB without obstacles. Moreover, the freights of two SLs need to be quoted on the blockchain distributed ledger. Further, SLA and SLB will experience multiple rounds of quotation.

There will be a fixed proportional relationship between freight $p_1$ of SLA and freight $p_2$ of SLB. The joint quotation strategy of the Freight Rate Alliance is illustrated in Figure 1. Two SLs reach an agreement on the quotation strategy in the consensus mechanism. Additionally, the relative freight $k$ is introduced to denote the fixed proportional relationship between two SLs. That is, SLA and SLB quote in fixed proportion. In other words, the freights of both SLs are adjusted at the same time and in the same range to maintain the stability of the relative freight. As a result, the lagged terms of the freights $\left[ p_1^n, p_2^m \right]$ become direct reference for both SLs. Additionally, the target under the joint quotation strategy is to find the optimal combination $\left[ p_1^{CQ}, k \right]$ to obtain the maximum profit of the Freight Rate Alliance.

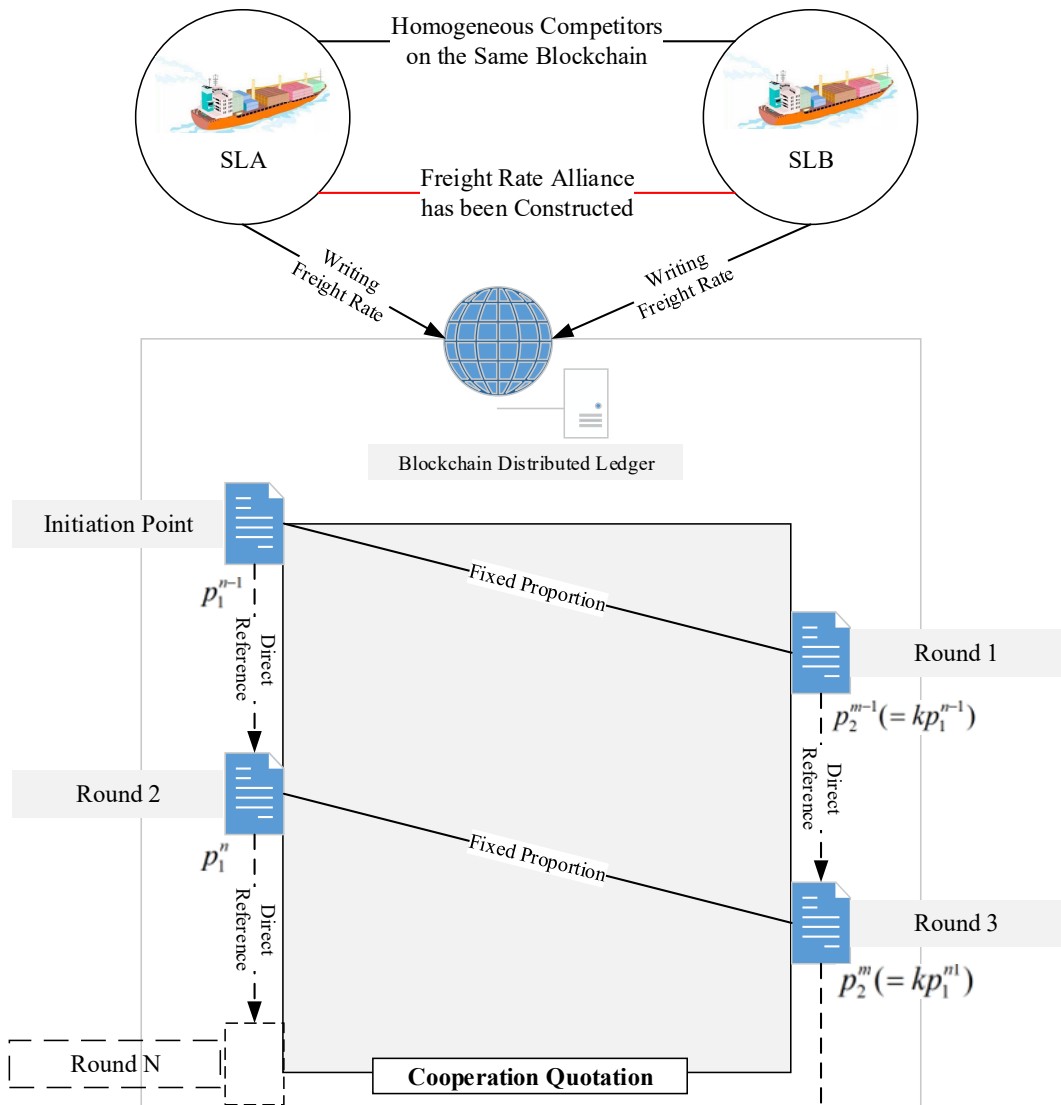

**Figure 1.** Multi-round joint quotation strategy of the Freight Rate Alliance.

### 3.1. Basic Assumptions

(1) The SLA and SLB are all in the same container shipping blockchain as the members. That means, quoting of freight is necessary for both SLs as members in accordance with the consensus on blockchain.

(2) Both SLs involved in this paper are all rational decision-makers. They have the same goals of improving their operation profit by healthy competition and cooperation on blockchain and avoiding the risk of a vicious price war. Additionally, the freight change is a pure rational behavior of SLs regardless of its impact on goodwill and the SLs' operation.

(3) It is assumed that both SLs can provide container shipping services from the same origin to the same destination on the same route. The impact of container shipping service diversity on the decision-making of SLs are not considered in this paper according to Zheng et al. [43].

(4) During the research period, the unit cost of each SL remains unchanged, because, based on the assumption of homogeneous service, unit cost reflects the scale effects of both SLs. Under the context of the transportation overcapacity and the market risk, the SLs with heavy asset characteristics will not easily adjust the asset allocation in the short term to prevent breaking the scale effect.

(5) The strategy choice whether to cooperate or not is based on whether they can improve their profits. The key factor affecting is excess profit in accordance with the cooperative game theory [44–46].

(6) The relationship between container shipping demand and freight essentially meets a linear relationship. Additionally, container shipping demand can be constructed as a multivariate linear function of $p_1$ and $p_2$ [47,48].

The actual meanings of parameters involved in this paper can be seen in Table 1.

**Table 1.** The actual meanings of the parameters involved.

| Parameters | Actual Meanings | Units |
|:---:|:---:|:---:|
| $\xi$ | The total demand forecast of container shipping blockchain market | TEU |
| $\theta_1$ | The expected market demand share of SLA | TEU |
| $\theta_2$ | The expected market demand share of SLB | TEU |
| $\alpha_1$ | The freight sensitivity coefficient for SLA | TEU/USD (TEU per dollar) |
| $\alpha_2$ | The freight sensitivity coefficient for SLB | TEU/USD (TEU per dollar) |
| $p_1$ | The freight of SLA | USD |
| $p_2$ | The freight of SLB | USD |
| $p_0$ | The benchmark freight of container shipping blockchain market | USD |
| $\gamma_1$ | The competition intensity of SLA from SLB | TEU/USD (TEU per dollar) |
| $\gamma_2$ | The competition intensity of SLB from SLA | TEU/USD (TEU per dollar) |
| $c_1$ | The unit cost of SLA | USD |
| $c_2$ | The unit cost of SLB | USD |
| $D_1$ | The container shipping demand of SLA | TEU |
| $D_2$ | The container shipping demand of SLB | TEU |
| $\pi_1$ | The profit of SLA | USD |
| $\pi_2$ | The profit of SLB | USD |
| $U_1$ | The utility of SLA | USD |
| $U_2$ | The utility of SLB | USD |

All of the parameters are over zero. The following is the relative relationship of some parameters.

(1) Because SLA is larger than SLB. $\theta_1 > \theta_2$, and $\theta_1 + \theta_2 = 1$.

(2) The freight sensitivity of the larger one is lower because of the scale effect and the long-term characteristics of the demand. That means, $\alpha_1 < \alpha_2$.

(3) The substitution effect from SLA will be stronger than that from SLB. So, $\gamma_1 < \gamma_2$.

(4) Because of the characteristics of the oligarchy and the long-term demand, the impact of substitution effect of shipping services on demand will be significantly lower than the freight. So, $\gamma_1 < \gamma_2 < \alpha_1 < \alpha_2$.

## 3.2. Fundamental Model

Competition in container shipping blockchain market with disclosure of freights is more intense than it is in the traditional market. So, the behavior of competitors plays an important role in the process of SL decision making. The effect of competitor's decision

making will be manifested as the substitution effect on the decision-makers' service, container shipping service in this paper. Additionally, the service substitution effect can be quantitatively expressed by competition intensity coefficient $\gamma_i$ ($i = 1, 2$). The competition intensity coefficient $\gamma_i$ represents the influence of competitor's freight on the SL's container shipping demand. The demand function [47,48] in this paper is defined as:

$$D_i(p_1, p_2) = \theta_i \xi - \alpha_i(p_i - p_0) + \gamma_i(p_j - p_i), i, j = 1, 2, i \neq j \tag{1}$$

In this way, the utility function of SLA and SLB can be obtained as:

$$U_i = \max \pi_i = \max[(p_i - c_i)D_i(p_1, p_2)] \, i = 1, 2 \tag{2}$$

To simplify the derivation and proof process, the fundamental model can be modified by some alternative parameters, including $d_i$ and $a_i$. Firstly, the basic demand function needs to be transformed into the following:

$$D_i(p_1, p_2) = \theta_i \xi + \alpha_i p_0 - (\alpha_i + \gamma_i)p_i + \gamma_i p_j \, i, j = 1, 2 \, i \neq j \tag{3}$$

As mentioned above, $d_i$ and $a_i$ are introduced. We obtained $d_i = \theta_i \xi + \alpha_i p_0, i = 1, 2$ and $a_i = \alpha_i + \gamma_i \, i = 1, 2$. The modified demand function can be obtained as:

$$D_i(p_1, p_2) = d_i - a_i p_i + \gamma_i p_j \, i, j = 1, 2 \, i \neq j \tag{4}$$

According to the fundamental assumptions, the alternative parameter $d_i$ refers to the constant in the multivariate linear demand function. Additionally, the alternative parameter $a_i$ refers to the actual competition intensity coefficient. The fundamental form of the demand function concentrates more on the factors influencing demand, while the modified form tends to highlight the impact of competitor's freight.

Although the decision-making process of the joint quotation is dynamic, the research in this part focuses more on the field of alliance profit maximization and the profit redistribution for members. So, the equivalent method and the thought of cooperative game theory were introduced.

Furthermore, in order to explore whether the joint quotation strategy can help container shipping lines to stabilize the profit level and achieve excess profit, it is necessary to build the initial theoretical equilibrium strategy of both shipping lines in an emerging shipping blockchain. Combined with the essential characteristics of the shipping blockchain alliance, the disclosure of freight information makes the container shipping market much closer to a perfectly competitive market. For an emerging shipping blockchain market that has reached a consensus on joint quotation, when the proportion of joint quotation is determined, its own optimal strategy also means the optimal strategy of the competitor. In the long run, although the decision-making process is dynamic, the equilibrium strategy will not change significantly when the number of decision-makers remains unchanged. Therefore, Cournot game model is more appropriate to construct the initial strategy combination of two shipping lines.

Without considering the decision-making order of two SLs in an emerging container shipping blockchain, the first-order optimal conditions for maximizing the profits of two SLs are constructed, respectively. First of all, the first derivative expressions of the two shipping lines' profits with respect to freights is as follows.

$$\begin{cases} \frac{\partial \pi_1}{\partial p_1} = d_1 - a_1 p_1 - a_1(p_1 - c_1) + p_2 \gamma_1 \\ \frac{\partial \pi_2}{\partial p_2} = d_2 - a_2 p_2 - a_2(p_2 - c_2) + p_1 \gamma_2 \end{cases} \tag{5}$$

Additionally, then, the first-order optimal conditions of both shipping lines can be obtained. Furthermore, the static equilibrium freight expressions of two SLs based on Cournot game can be obtained by solving the two first-order optimal conditions.

$$\begin{cases} p_1 = \frac{2a_1a_2c_1 + 2a_2d_1 + a_2c_2\gamma_1 + d_2\gamma_1}{4a_1a_2 - \gamma_1\gamma_2} \\ p_2 = \frac{2a_1a_2c_2 + 2a_1d_2 + a_1c_1\gamma_2 + d_1\gamma_2}{4a_1a_2 - \gamma_1\gamma_2} \end{cases} \tag{6}$$

Therefore, expressions of the reference freight and the reference profit level under the initial static equilibrium strategy can be obtained by substituting the parameter expressions.

*3.3. Model Analysis*

Under the joint quotation strategy, the two SLs make decisions by the method of equal proportional adjustment of the freight. The demand functions of both SLs are still the same as the fundamental model. The relationship between two freights under the joint quotation strategy can be expressed as: $p_2 = kp_1$ in accordance with the agreement. Hence, freight $p_1$ and freight $p_2$ are changed by the same margin when both SLs make an adjustment to their freights. The demand functions after iterating of SLA and SLB can be expressed as:

$$\begin{cases} D_1(p_1, k) = d_1 - a_1p_1 + \gamma_1kp_1 \\ D_2(p_1, k) = d_2 - a_2kp_1 + \gamma_2p_1 \end{cases} \tag{7}$$

$D_1(p_1, k)$ and $D_2(p_1, k)$ are separately container shipping demands of SLA and SLB. $d_1$ and $d_2$ separately refer to the constant parameter terms of SLA and SLB. $a_1$ and $a_2$ separately refer to the actual effect of the freights on the demands. $\gamma_1$ and $\gamma_2$ indicate the effect of the competitor on the container shipping demand. Because SLB is the smaller one, its freight cannot over the larger one, SLA. Only doing so, can SLB obtain the container shipping demand. Therefore, $k \leq 1$. Furthermore, the relative freight $k$ needs to satisfy the going concern assumptions of SLB. That means, $kp_1 - c_2 > 0$. So, the value range of $k$ is

$$\frac{c_2}{p_1} < k \leq 1.$$

The utility functions of SLA and SLB under the joint quotation strategy is consistent with the fundamental model.

$$\begin{cases} U_1 = \max\pi_1 = \max[(p_1 - c_1)D_1(p_1, k)] \\ U_2 = \max\pi_2 = \max[(kp_1 - c_2)D_2(p_1, k)] \end{cases} \tag{8}$$

Therefore, the total utility function can be obtained as:

$$U_T = \max\pi_T = \max[(p_1 - c_1)D_1(p_1, k) + (kp_1 - c_2)D_2(p_1, k)] \tag{9}$$

Obviously, the total utility function is the binary function on the freight $p_1$ and the relative freight $k$. Separately find the partial derivatives of the function on both variables. The expressions can be obtained as:

$$\begin{cases} \frac{\partial\pi_1}{\partial p_1} = k[a_2c_2 + d_2 + p_1\gamma_1 + \gamma_1(p_1 - c_1) + 2p_1\gamma_2] + d_1 - a_1p_1 \\ \qquad -2k^2a_2p_1 - a_1(p_1 - c_1) - c_2\gamma_2 \\ \frac{\partial\pi_2}{\partial k} = a_2c_2p_1 + d_2p_1 - 2ka_2p_1^2 + p_1(-c_1 + p_1)\gamma_1 + p_1^2\gamma_2 \end{cases} \tag{10}$$

The optimal freight combination should appear at the point of the maximum total profit can be achieved. So, $\frac{\partial\pi_1}{\partial p_1}$ and $\frac{\partial\pi_2}{\partial k}$ are equal to zero, respectively. Additionally, the expressions of the strategy set $[p_1^*, k^*]$ can be obtained by solving the equations.

$$\begin{cases} p_1^* = \dfrac{2a_1a_2c_1 + 2a_2d_1 + a_2c_2\gamma_1 + d_2\gamma_1 + d_2\gamma_2 - c_1\gamma_1^2 - a_2c_2\gamma_2 - c_1\gamma_1\gamma_2}{4a_1a_2 - (\gamma_1 + \gamma_2)^2} \\ k^* = \dfrac{2a_1a_2c_2 + 2a_1d_2 + a_1c_1\gamma_2 + d_1\gamma_1 + d_1\gamma_2 - c_2\gamma_1\gamma_2 - a_1c_1\gamma_1 - c_2\gamma_2^2}{2a_1a_2c_1 + 2a_2d_1 + a_2c_2\gamma_1 + d_2\gamma_1 + d_2\gamma_2 - c_1\gamma_1\gamma_2 - a_2c_2\gamma_2 - c_1\gamma_1^2} \end{cases} \tag{11}$$

Based on the solution set $\left[ p_1^*, k^* \right]$, the total profit can be solved out as:

$$\begin{aligned} \pi_T^* = \dfrac{1}{4a_1a_2 - (\gamma_1 + \gamma_2)^2} \big[ & a_1^2 a_2 c_1^2 + a_1 a_2^2 c_2^2 - 2a_1 a_2 c_2 d_2 + (\gamma_1 + \gamma_2) * \\ & (d_1 + c_2\gamma_1)(d_2 + c_1\gamma_2) + a_2(d_1 + c_2\gamma_1)(d_1 - c_2\gamma_2) + (d_2 - c_1\gamma_1) * \\ & a_1(d_2 + c_1\gamma_2) - 2c_1 a_1 a_2 d_1 - c_2 c_1 a_1 a_2 (\gamma_1 + \gamma_2) \big] \end{aligned} \tag{12}$$

Until now, there are several key problems need to be verified, including the existence and the rationality of the solution set $\left[ p_1^*, k^* \right]$.

**Lemma 1.** $d_i > c_i a_i$ *is always true for any* $a_i, c_i, d_i > 0$, $i = 1, 2$ *on the basis of all parameters over zero.*

**Proof.** Taking SLA as an example.

Substitute the expressions $d_1 = \xi\theta_1 + \alpha_1 p_0$ and $a_1 = \alpha_1 + \gamma_1$ into SLA's demand function. The demand function can be replaced as: $D_1(p_1, p_2) = d_1 - a_1 p_1 + \gamma_1 p_2$.

Take SLA as an example. Firstly, change the form of SLA's iterated demand function of SLA. The new expression $a_1 = \dfrac{d_1 - D_1(p_1, p_2) + \gamma_1 p_2}{p_1}$ can be obtained. Obviously, $\dfrac{d_1 - D_1(p_1, p_2) + \gamma_1 p_2}{p_1} > 0$ is true because $a_i i = 1, 2$ is positive. Additionally, $d_1 - D_1(p_1, p_2) + \gamma_1 p_2 > 0$ is true. So,

$$\dfrac{d_1 - D_1(p_1, p_2) + \gamma_1 p_2}{c_1} > \dfrac{d_1 - D_1(p_1, p_2) + \gamma_1 p_2}{p_1}$$

is true according to the practical significance $p_i > c_i, i = 1, 2$. Theoretically, container shipping industry is dominated by long-term demand. So, freight is the main factor rather than the substitution effect affecting the container shipping demand. In this way, $D_1(p_1, p_2) - \gamma_1 p_2 > 0$ is true. Additionally, the following relationship will be true:

$$\dfrac{d_1}{c_1} > \dfrac{d_1 - D_1(p_1, p_2) + \gamma_1 p_2}{c_1} > \dfrac{d_1 - D_1(p_1, p_2) + \gamma_1 p_2}{p_1}$$

So, $\dfrac{d_1}{c_1} > a_1$, $d_1 > c_1 a_1$. The proof to SLB is completely consistent with SLA. □

**Lemma 2.** $c_1 \left( a_1 a_2 - \gamma_1^2 \right) > 0$ *is true based on the assumptions of all parameters positive.*

**Proof.** According to the substitutions of the intermediate parameters $\begin{cases} a_1 = \gamma_1 + \alpha_1 \\ a_2 = \gamma_2 + \alpha_2 \end{cases}$, $a_1, a_2 > \gamma_1, \gamma_2$

So, $a_1 a_2 > \gamma_1^2$. $c_1 > 0$ is true based on the fundamental assumptions of all parameters involved over zero. Therefore,

$$c_1 \left( a_1 a_2 - \gamma_1^2 \right) > 0$$

□

**Lemma 3.** $\gamma_1 (a_2 c_2 - c_1 \gamma_2) > 0$ *is true based on the assumptions of all parameters positive.*

**Proof.** Based on the substitutions of the intermediate parameters, $a_2 > \gamma_2$. Additionally, according to the fundamental assumptions, SLA is larger than SLB. So, the scale economy

of SLA is significantly stronger than that of SLB. In other words, SLA can achieve lower unit cost than SLB, $c_1 < c_2$. Therefore,

$$a_2c_2 - c_1\gamma_2 > 0$$

Further, $\gamma_1 > 0$ is true in accordance with the assumptions of all parameters positive and its practical significance. So, $\gamma_1(a_2c_2 - c_1\gamma_2) > 0$ is true. □

**Lemma 4.** $p_1^* > 0$ *is always true based on the assumptions of all parameters positive.*

**Proof.** Consider the numerator and the denominator of the expression of $p_1^*$.

For the numerator $2a_1a_2c_1 + 2a_2d_1 + a_2c_2\gamma_1 + d_2\gamma_1 + d_2\gamma_2 - c_1\gamma_1^2 - a_2c_2\gamma_2 - c_1\gamma_1\gamma_2$, there are three inequality relations need to be verified, including $a_1a_2c_1 - c_1\gamma_1^2$, $d_2\gamma_2 - a_2c_2\gamma_2$ and $a_2c_2\gamma_1 - c_1\gamma_1\gamma_2$. Factoring the above expressions as: $c_1(a_1a_2 - \gamma_1^2)$, $\gamma_2(d_2 - a_2c_2)$ and $\gamma_1(a_2c_2 - c_1\gamma_2)$. All of the three expressions are positive separately according to Lemmas 1–3. So, the numerator of $p_1^*$ is positive.

For the denominator $4a_1a_2 - (\gamma_1 + \gamma_2)^2$, according to the substitutions of the intermediate parameters, $a_1a_2 = (\alpha_1 + \gamma_1)(\alpha_2 + \gamma_2)$. As the container shipping market depends on long-term transportation contract, the long-term demand is dominant. According to the relative relationship among parameters, $\alpha_1, \alpha_2 > \gamma_1, \gamma_2$. So,

$$(\alpha_1 + \gamma_1)(\alpha_2 + \gamma_2) > (\gamma_1 + \gamma_2)^2$$

$$4a_1a_2 - (\gamma_1 + \gamma_2)^2 > 0$$

Above all, both the numerator and the denominator of $p_1^*$ are positive, $p_1^* > 0$ is true. □

**Proposition 1.** *There is a reasonable and practical $p_1^*$, which can maximize the profit of the Freight Rate Alliance.*

**Proof.** Based on Lemma 4, there is a positive solution $p_1^*$ can help to achieve the maximum value of the function $\pi_T(p_1, k)$. However, the practicality of $p_1^*$ refers to the equilibrium freight which needs to satisfy the assumptions of normal going concern of both SLs. That is, $p_1^*$ needed to be over $c_1$. Subtract $c_1$ from $p_1^*$, the following results can be obtained.

$$p_1^* - c_1 = \frac{2a_2(d_1 - a_1c_1) + \gamma_2(d_2 - a_2c_2) + a_2c_2\gamma_1 + d_2\gamma_1 + c_1\gamma_1\gamma_2 + c_1\gamma_2^2}{4a_1a_2 - (\gamma_1 + \gamma_2)^2}$$

The denominator of the above expression $4a_1a_2 - (\gamma_1 + \gamma_2)^2$ is positive referring to Lemma 4. Consider the numerator $2a_2(d_1 - a_1c_1) + \gamma_2(d_2 - a_2c_2) + a_2c_2\gamma_1 + d_2\gamma_1 + c_1\gamma_1\gamma_2 + c_1\gamma_2^2$. Obviously, the numerator is positive on the basis of all parameters involved positive according to Lemma 1, $d_i > a_ic_i, i = 1, 2$. Therefore, $p_1^* - c_1 > 0$

Above all, the equilibrium freight $p_1^*$ is reasonable and with practical significance. □

**Lemma 5.** *There exists the following relationship between the substitution parameters $d_1$ and $d_2$, $d_1 \geq d_2$.*

**Proof.** Consider the substituted demand functions of both SLs as follows:

$$\begin{cases} D_1(p_1, p_2) = d_1 - a_1p_1 + \gamma_1p_2 \\ D_2(p_1, p_2) = d_2 - a_2p_2 + \gamma_2p_1 \end{cases}$$

Because the scale of SLA is larger than that of SLB, if the freight of SLA equals to that of SLB, the container shipping demand of SLA will be more than that of SLB because of the greater influence on the container shipping traditional market of SLA without considering the service differentiation. Thus, making $p_1$ equal to $p_2$, the demand functions of both SLs will be as follows.

$$\begin{cases} D_1(p_1, p_2) = d_1 - \alpha_1 p_1 \\ D_2(p_1, p_2) = d_2 - \alpha_2 p_2 \end{cases}$$

For $D_1(p_1, p_2)$ is more than $D_2(p_1, p_2)$,

$$\begin{cases} d_1 - \alpha_1 p_1 > d_2 - \alpha_2 p_2 \\ p_1 = p_2 \end{cases}$$

Solve the inequation. Then, the relationship between the freight $p_1$ and the parameters involved are obtained.

$$p_1 > \frac{d_1 - d_2}{\alpha_1 - \alpha_2}$$

Something interesting is that the strong market influence of SLA always exists, which has nothing to do with whether SLA runs at a loss. That means, the inequality relationship will exist as long as $p_1$ exists. So, $p_1 > 0$ needs to be true combined with its practical significance. So,

$$\frac{d_1 - d_2}{\alpha_1 - \alpha_2} \leq 0$$

Due to $\alpha_1 < \alpha_2$ on the basis of SLA being larger and the long-term demand assumption, $d_1 - d_2 \geq 0$ needs to be true. Therefore, $d_1 \geq d_2$.

Only when the freight of SLB is lower than that of SLA, the strong market influence of SLA will be more prominent. □

**Proposition 2.** *There is a practical $k^*$ meeting assumption of going concern of SLB, which can maximize the profit of the Freight Rate Alliance. That means,*

$$k^* > \frac{c_2}{p_1}.$$

**Proof.** Prove the inequality $k^* - \frac{c_2}{p_1} > 0$.

$$k^* - \frac{c_2}{p_1} = \frac{(d_1 + c_2\gamma_1)(\gamma_1 + \gamma_2) + a_1[2d_2 - 2a_2c_2 + c_1(\gamma_2 - \gamma_1)]}{(d_2 - c_1\gamma_1)(\gamma_1 + \gamma_2) + a_2[2d_1 + 2a_1c_1 + c_2(\gamma_1 - \gamma_2)]}$$

According to Lemma 1, $d_2 - a_2c_2 > 0$. Because SLA's scale is larger than SLB based on the fundamental assumptions, SLA's substitution effect to SLB ($\gamma_2$) is stronger than SLB's substitution effect to SLA ($\gamma_1$). $\gamma_2 - \gamma_1 > 0$. Therefore, the numerator of the expression is positive.

Consider the denominators. For $d_2 - a_2c_2 > 0$, $a_1, a_2 > \gamma_1, \gamma_2$ and $c_2 > c_1$ on the basis of Lemma 1 and the fundamental assumptions, $a_2c_2 > c_1\gamma_1$.

So,

$$d_2 - c_1\gamma_1 > 0$$

For $d_1 > d_2$, $d_1 > a_2c_2$ according to Lemmas 1 and 5. In the same way, $a_2c_2 > c_2\gamma_2$. So,

$$d_1 - c_2\gamma_2 > 0$$

The denominator of the expression is also positive. Therefore,

$$k^* > \frac{c_2}{p_1}$$

Above all, the equilibrium relative freight $k^*$ is with practical significance by meeting the going concern assumptions about SLB. $\square$

**Proposition 3.** *The maximum value of the total profit exists when all parameters involved are positive.*

**Proof.** The total profit expression can be obtained as:

$$
\begin{aligned}
\pi_T^* = \frac{1}{4a_1a_2-(\gamma_1+\gamma_2)^2} \big[ & a_1^2 a_2 c_1^2 + (\gamma_1+\gamma_2)(d_1+c_2\gamma_1)(d_2+c_1\gamma_2) \\
& +a_2(d_1+c_2\gamma_1)(d_1-c_2\gamma_2) + a_1 a_2^2 c_2^2 + a_1(d_2-c_1\gamma_1)(d_2+c_1\gamma_2) - \\
& 2a_1a_2c_2d_2 - 2c_1a_1a_2d_1 - c_1a_1a_2c_2(\gamma_1+\gamma_2) \big]
\end{aligned}
$$

To simplify the derivation and proof process, the expression of $\pi_T^*$ can be made to be the following form. $Q$ represents the numerator expression of $\pi_T^*$. Additionally, $L$ represents the denominator expression of $\pi_T^*$.

$$
\pi_T^* = \frac{Q}{L}
$$

According to the Lemma 4, $L$ is positive. $Q$ is also positive on the basis of Lemma 6. Therefore, $\pi_T^*$ is over than zero when all parameters involved positive. So, the optimal total profit of both SLs exists. $\square$

**Lemma 6.** *The expression $Q$ is always positive only if the parameters involved are all positive.*

**Proof.** Assume $Q$ is made to be a function. Then, $Q$ will be a typical quadratic function on $d_1$, $d_2$, $c_1$ and $c_2$.

$$
\begin{aligned}
Q(d_1,d_2,c_1,c_2) = & a_1^2 a_2 c_1^2 + (\gamma_1+\gamma_2)(d_1+c_2\gamma_1)(d_2+c_1\gamma_2) + a_2(d_1+c_2\gamma_1)(d_1-c_2\gamma_2) + a_1 a_2^2 c_2^2 + \\
& a_1(d_2-c_1\gamma_1)(d_2+c_1\gamma_2) - 2a_1a_2c_2d_2 - 2c_1a_1a_2d_1 - c_1a_1a_2c_2(\gamma_1+\gamma_2)
\end{aligned}
$$

The highest order items of $d_1$, $d_2$, $c_1$ and $c_2$ are $a_2$, $a_1$, $\left(a_1^2 a_2 - a_1\gamma_1\gamma_2\right)$ and $\left(a_1 a_2^2 - a_2\gamma_1\gamma_2\right)$. On the basis of the fundamental assumptions of all parameters positive and Lemma 2, all of the four items are positive. That means, function $Q$ is a quadratic function with respect to the opening up of four variables, respectively. In accordance with the characteristics of quadratic function, in order to judge whether $Q$ is positive or negative, it is only needed to check if there is real root for each variable. So,

$$
Q_\Delta \begin{cases}
\Delta_{d_1} = -(d_2 - a_2c_2 + c_1\gamma_2)^2 \left[4a_1a_2 - (\gamma_1+\gamma_2)^2\right] \\
\Delta_{d_2} = -(d_1 - a_1c_1 + c_2\gamma_1)^2 \left[4a_1a_2 - (\gamma_1+\gamma_2)^2\right] \\
\Delta_{c_1} = -[a_1(a_2c_2 - d_2) - \gamma_2(d_1+c_2\gamma_1)]^2 \left[4a_1a_2 - (\gamma_1+\gamma_2)^2\right] \\
\Delta_{c_2} = -[a_2(d_1 - a_1c_1) + \gamma_1(d_2+c_1\gamma_2)]^2 \left[4a_1a_2 - (\gamma_1+\gamma_2)^2\right]
\end{cases}
$$

For $\Delta_{d_1}$, $4a_1a_2 - (\gamma_1+\gamma_2)^2$ is significantly positive according to Lemma 4. Additionally, $d_2 - a_2c_2 + c_1\gamma_2$ is also positive due to $d_2 > a_2c_2$ on the basis of Lemma 1. So, $\Delta_{d_1} < 0$. In the same way, $\Delta_{d_2} < 0$ can also be proved. Therefore, there is no real root for the function $Q$ on $d_1$ and $d_2$.

For $\Delta_{c_1}$, $4a_1a_2 - (\gamma_1+\gamma_2)^2$ is significantly positive according to Lemma 4. Additionally, $a_2c_2 - d_2 < 0$ on the basis of Lemma 1. $[a_1(a_2c_2 - d_2) - \gamma_2(d_1+c_2\gamma_1)]^2$ is significantly positive. So, $\Delta_{c_1} < 0$. In the same way, $\Delta_{c_2} < 0$ can also be proved. Therefore, there is no real root for the function $Q$ on $c_1$ and $c_2$.

Above all, $Q$ is quadratic function with opening up and without real root on $d_1$, $d_2$, $c_1$ and $c_2$. That is, $Q$ is always positive when parameters involved are all over zero. $\square$

## 4. Numerical Experiment

According to the research of the Model Development, the parameter expressions of the equilibrium solution under the joint quotation strategy can be obtained in the model analysis. Through the demonstration, it is concluded that the equilibrium solutions under the joint quotation strategy obtained based on the model in this paper are of practical significance. That is, both equilibrium freight rates exceed the unit costs of two container shipping lines. However, it cannot be obtained through model analysis whether the joint quotation strategy can make the SLs achieve high profits than the initial theoretical equilibrium strategies. So, it needs to be further analyzed by introducing numerical experiments. Furthermore, sensitivity analysis is adopted to study the effect of some important parameters on freights, container shipping demands and profits of SLA and SLB. The assignment of relevant parameters referring to [49,50] can be seen in Table 2.

**Table 2.** The assignment of relevant parameters.

| Parameters | Meaning and Description | SLA (The Larger Shipping Line) ($i = 1$) | SLB (The Smaller Shipping Line) ($i = 2$) | Parameter and Variable Units |
|---|---|---|---|---|
| $\theta_i$ | The expected market demand share of the shipping lines. ($\theta_1$ refers to the expected market demand share of SLA. $\theta_2$ refers to the expected market demand share of SLB.) | 0.6 | 0.4 | % |
| $\alpha_i$ | The nominal freight sensitivity coefficient for the shipping line. ($\alpha_1$ refers to the nominal freight sensitivity coefficient for SLA. $\alpha_2$ refers to the nominal freight sensitivity coefficient for SLB.) | 30 | 35 | TEU/USD (TEU per dollar) |
| $\gamma_i$ | The competition intensity of the shipping line. ($\gamma_1$ refers to the competition intensity of SLA from SLB. $\gamma_2$ refers to the competition intensity of SLB from SLA.) | 10 | 15 | TEU/USD (TEU per dollar) |
| $c_i$ | The unit cost of the shipping line ($c_1$ refers to the unit cost of SLA. $c_2$ refers to the unit cost of SLB.) | 350 | 600 | USD (dollar) |
| $p_0$ | The benchmark freight of container shipping blockchain market. | 1000 | | USD (dollar) |
| $\xi$ | The total demand forecast of container shipping blockchain market. | 80,000 | | TEU |
| $a_i$ | According to the parameter substitution relationship, for SLA $a_1 = \alpha_1 + \gamma_1$, and for SLB $a_2 = \alpha_2 + \gamma_2$. The actual freight sensitivity coefficient for the shipping line. ($a_1$ refers to the actual freight sensitivity coefficient for SLA. $a_2$ refers to the actual freight sensitivity coefficient for SLB.) | 40 | 50 | TEU/USD (TEU per dollar) |

**Table 2.** *Cont.*

| Parameters | Meaning and Description | SLA (The Larger Shipping Line) (*i* = 1) | SLB (The Smaller Shipping Line) (*i* = 2) | Parameter and Variable Units |
|---|---|---|---|---|
| $d_i$ | According to the parameter substitution relationship, for SLA $d_1 = \xi\theta_1 + \alpha_1 p_0$, and for SLB $d_2 = \xi\theta_2 + \alpha_2 p_0$ The actual market share for the shipping line. ($d_1$ refers to the actual market share for SLA. $d_2$ refers to the actual market share for SLB.) | 78,000 | 67,000 | TEU/USD (TEU per dollar) |

### 4.1. Numerical Experiment Result

The numerical experiment result can be obtained as the follows based on the above parameter assignment method. In order to make a more accurate comparison of freight, container shipping demand and profit between two strategies, the difference ratio $\delta$ is introduced into this paper. The definition of $\delta$ is as:

$$\delta = \frac{S_2 - S_1}{S_1} \tag{13}$$

where $\delta$ refers to the size of the relative difference between two states. Additionally, $\delta$ can be positive or negative. The positive $\delta$ means the same variable under state two is larger than under state one. On the contrary, the negative $\delta$ means the same variable under state two is smaller than under state one. $S_1$ and $S_2$ refer to the values of the same variable in different states. $S_1$ refers to the Initial Theoretical Equilibrium state. $S_2$ refers to the blockchain state with joint quotation strategy.

According to Table 3, the profit difference ratio of SLA and SLB under two strategies are 1.56% and 3.20%. Additionally, the difference ratio of total profit is 2.07%. Compared with the Initial Theoretical Equilibrium strategy, the joint quotation can help the two SLs to achieve excess profits. Additionally, the impact of the Joint quotation mode on SLB's profit is more significant than that on SLA's profit. Additionally, the total profit under the joint quotation strategy is higher than that under the Initial Theoretical Equilibrium strategy. Consider the freights under the joint quotation strategy. The difference ratio of both SLs between two strategies are 9.74% of SLA and 11.05% of SLB. The demand difference ratio of both SLs is separately SLA −10.24% and SLB −15.85%. That means, the container shipping demand level of both SLs will suffer from a significant decrease. Additionally, the impact of the joint quotation mode on SLB's container shipping demand is more significant than that on SLA's container shipping demand. The total container shipping demand difference ratio is −12.82%.

**Table 3.** The numerical experiment results.

| Decision Maker | Initial Theoretical Equilibrium Strategy | | Joint Quotation Strategy | |
|---|---|---|---|---|
| | SLA | SLB | SLA | SLB |
| Freight (USD) | 1314 | 1167 | 1442 | 1296 |
| Demand (1000 TEU) | 37.1 | 28.4 | 33.3 | 23.9 |
| Total Demand (1000 TEU) | 65.5 | | 57.1 | |
| Profit (1000 USD) | 35,775 | 16,080 | 36,333 | 16,594 |
| Total Profit (1000 USD) | 51,855 | | 52,928 | |

The numerical experiment result is interesting. The joint quotation strategy can help both SLA and SLB to obtain excess profit, because the way to maximize profit the under joint quotation strategy is to greatly improve the freights. The cost of maximizing profit is to sacrifice part of the container shipping demand. However, the difference ratios of profit between the joint quotation strategy and the Initial Theoretical Equilibrium strategy are much smaller than the difference ratios of container shipping demand between two strategies. That means, with substantial increases in freights and substantial decreases in container shipping demand, only very limited increases in profits can be achieved under the joint quotation strategy compared with the Initial Theoretical Equilibrium strategy.

In order to compare two SLs' profits under the joint quotation strategy with the Initial Theoretical Equilibrium strategy, the three-dimensional images covering profits of SLA and SLB are as follows.

As illustrated in Figure 2, the curve $\pi_T^{JQ}$ represents the total profit under the joint quotation strategy and the curve $\pi_T^I$ represents the total profit under the Initial Theoretical Equilibrium strategy. It is very possible for SLA and SLB to obtain excess profits by adopting the joint quotation strategy, compared to the Initial Theoretical Equilibrium strategy. Only if the decision variables $p_1$ and $k$ under the joint quotation strategy satisfy the relative relationship as:

$$\begin{cases} k > \dfrac{0.25(3740.+p_1)}{p_1} - 0.001087624927695377\sqrt{\dfrac{-1.278920512544\times10^{12}+1.7985034\times10^9 p_1 - 623453. p_1^2}{p_1^2}} \\ k < \dfrac{0.25(3740.+p_1)}{p_1} + 0.001087624927695377\sqrt{\dfrac{-1.278920512544\times10^{12}+1.7985034\times10^9 p_1 - 623453. p_1^2}{p_1^2}} \end{cases}$$

Additionally, on the closed surface, the value range of $p_1$ need to be $1271.81 < p_1 < 1612.93$. Based on the relative relationship and the value range of $p_1$, the value range of $k$ can be obtained as $0.83 < k < 0.99$. So, the condition one $C_1$ to achieve excess total profit is

$$\left[ \begin{array}{l} 1271.81 < p_1 < 1612.93 \\ \&\& \begin{cases} k > \dfrac{0.25(3740.+p_1)}{p_1} - 0.001087624927695377\sqrt{\dfrac{-1.278920512544\times10^{12}+1.7985034\times10^9 p_1 - 623453. p_1^2}{p_1^2}} \\ k < \dfrac{0.25(3740.+p_1)}{p_1} + 0.001087624927695377\sqrt{\dfrac{-1.278920512544\times10^{12}+1.7985034\times10^9 p_1 - 623453. p_1^2}{p_1^2}} \end{cases} \end{array} \right].$$

The optimal freight rates under the joint quotation strategy that can realize excess profit will exist in the space $[1271.81 < p_1 < 1612.93 \&\& 0.83 < k < 0.99]$.

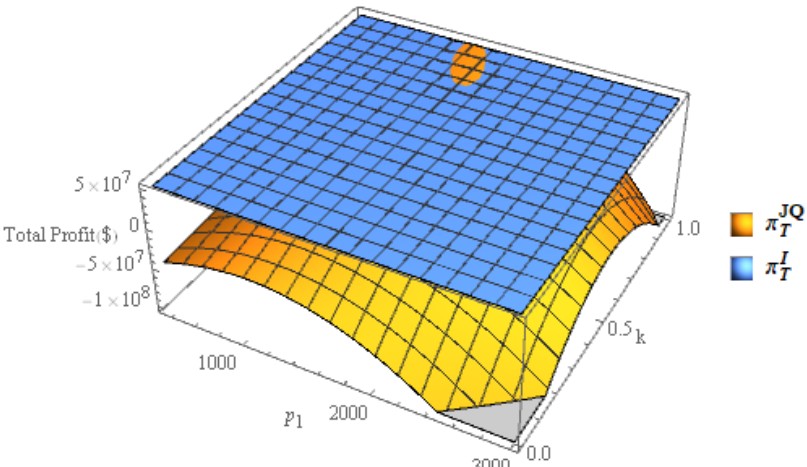

**Figure 2.** Total profit curves.

Furthermore, both of the two SLs also need to achieve excess profits, otherwise the cooperation cannot be maintained. Thus, both SLs' profit surfaces are obtained. It is clearly

that both SLA and SLB can achieve excess profit by adopting the joint quotation strategy according to Figures 3 and 4.

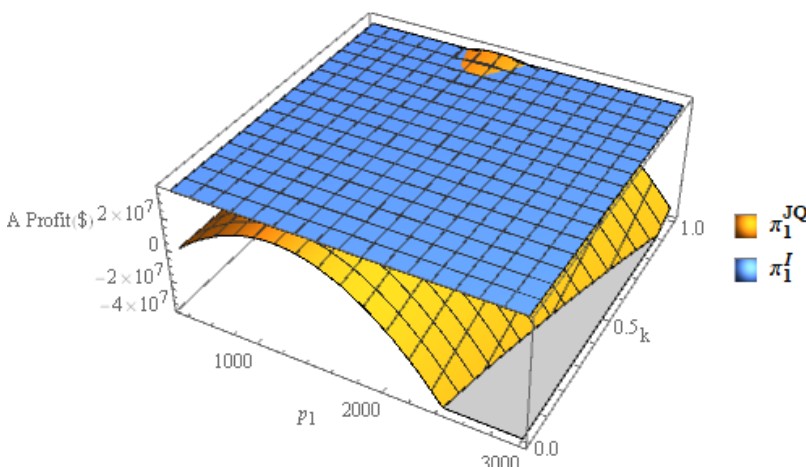

**Figure 3.** SLA's profit curves.

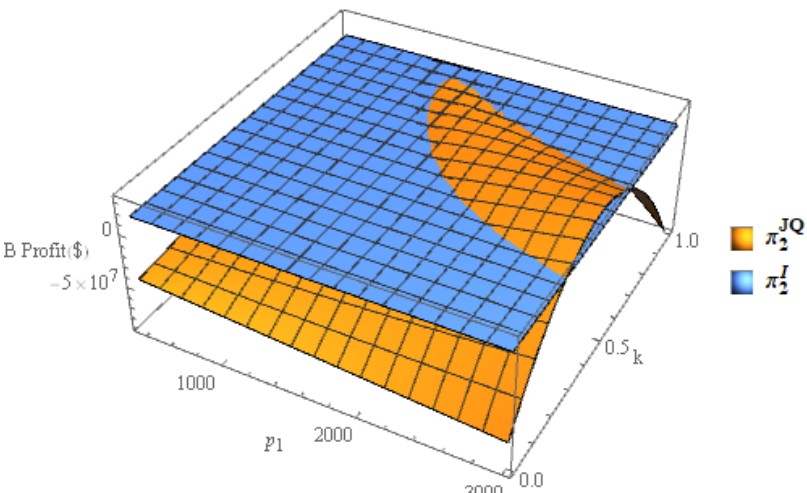

**Figure 4.** SLB's profit curves under two strategies.

As illustrated in Figure 3, the curve $\pi_1^{JQ}$ represents SLA's profit under the joint quotation strategy and the curve $\pi_1^I$ represents SLA's profit under the Initial Theoretical Equilibrium strategy. Aiming to achieve the excess profit of SLA under the Joint Quotation strategy, it is needed to be satisfied with the relative relationship for the decision variables $p_1$ and $k$ as:

$$k > \frac{98596.p_1^2 - 2.267708 \times 10^8 p_1 + 1.5544138 \times 10^{11}}{24649.p_1^2 - 8627150.0 p_1}$$

Additionally, on the surface, the value ranges of $p_1$ need to be $p_1 > 350$. Based on the relative relationship and the value range of $p_1$, the value range of $k$ can be obtained as $0.86 < k \leq 1$. So, the condition two $C_2$ to achieve excess total profit is

$$\left[ p_1 > 350 \&\& k > \frac{98596.p_1^2 - 2.267708 \times 10^8 p_1 + 1.5544138 \times 10^{11}}{24649.p_1^2 - 8627150.0 p_1} \right]$$

The optimal freight rates under the joint quotation strategy that can realize excess profit will exist in the space $[p_1 > 350 \& 0.69 < k \leq 1]$.

As illustrated in Figure 4, the curve $\pi_2^{JQ}$ represents SLB's profit under the joint quotation strategy and the curve $\pi_2^I$ represents SLB's profit under the Initial Theoretical Equilib-

rium strategy. In order to achieve the excess profit of SLB under the joint quotation strategy, the decision variables $p_1$ and $k$ need to satisfy the relative relationship as:

$$\begin{cases} k > \dfrac{0.05(19400.+3.p_1)}{p_1} - 0.0008092281863445159\sqrt{\dfrac{34359.p_1^2+1.695044\times10^8 p_1-2.82042771544\times10^{11}}{p_1^2}} \\ k < \dfrac{0.05(19400.+3.p_1)}{p_1} + 0.0008092281863445159\sqrt{\dfrac{34359.p_1^2+1.695044\times10^8 p_1-2.82042771544\times10^{11}}{p_1^2}} \end{cases}$$

Additionally, on the surface, the value ranges of $k$ need to be $0 < k < 0.89$. Based on the relative relationship and the value range of $k$, the value range of $p_1$ can be obtained as $p_1 > 1313.96$. So, the condition three $C_3$ to achieve excess total profit is

$$\left[ \begin{array}{l} p_1 > 1313.96 \\ \&\& \begin{cases} k > \dfrac{0.05(19400.+3.p_1)}{p_1} - 0.0008092281863445159\sqrt{\dfrac{34359.p_1^2+1.695044\times10^8 p_1-2.82042771544\times10^{11}}{p_1^2}} \\ k < \dfrac{0.05(19400.+3.p_1)}{p_1} + 0.0008092281863445159\sqrt{\dfrac{34359.p_1^2+1.695044\times10^8 p_1-2.82042771544\times10^{11}}{p_1^2}} \end{cases} \end{array} \right].$$

The optimal freight rates under the joint quotation strategy that can realize excess profit will exist in the space $[p_1 > 1313.96 \,\&\&\, 0 < k < 0.89]$.

Therefore, the decision space $C_T$ can not only realize the total excess profit, but also ensure that each SL can obtain excess profit. $C_T$ can be obtained by taking the intersection of decision variables $p_1$ and $k$ in $C_1$, $C_2$ and $C_3$ three subspaces. So, $C_T$ is $[1313.96 < p_1 < 1612.93 \,\&\, 0.83 < k < 0.89]$. Only in the decision space $C_T$ can both SLs obtain excess profits and excess total profit. That means that the joint quotation strategy has practical significance.

### 4.2. Sensitivity Analysis

For further investigating the impact of the internal competition between members of the shipping blockchain alliance on freight, profit and container shipping demand, a sensitivity analysis was introduced in this paper. The competition intensity coefficients are key parameters characterizing the internal competition relationship in the shipping blockchain alliance. Therefore, the impact of $\gamma_1$ and $\gamma_2$ on freights, profits and demands of two container shipping lines is the main focus of this section.

#### 4.2.1. The Relative Freight $k$ Sensitivities on the Competition Intensity Coefficients

The effects of $\gamma_1$ and $\gamma_2$ on the relative freight of SLB under the joint quotation strategy are the focus of this section, because the variable $k$ directly determines the long-term sustainability of the container shipping blockchain.

In accordance with Figure 5, when the service substitution effect of SLB increases, higher freight can be adopted by SLB. This is a very good signal for both SLA and SLB. For SLB, the profit space can be improved as the freight between two SLs narrows. In other words, the freight of SLB tends to be more reasonable. For SLA, more container shipping demand will be attracted thanks to the freight improvement of SLB's freight.

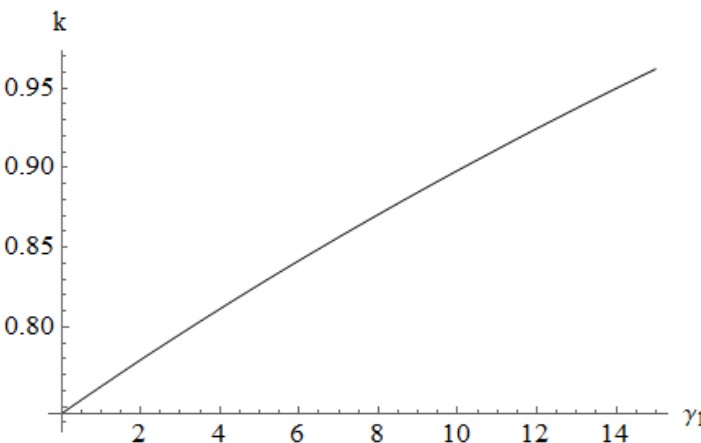

**Figure 5.** The relative freight sensitivity of $\gamma_1$.

According to Figure 6, when the service substitution effect of SLA increases, SLB needs to reduce the relative freight $k$ in order to maintain the maximum profit of the alliance and obtain excess profit. SLB needs to reduce the relative freight $k$ in order to maintain the maximum profit of the alliance and obtain excess profit. In other words, SLB needs to adopt a much lower freight compared to SLA to maintain the stability. Although the enhancement of SLA's service substitution effect will lead to the decline of the relative freight, the decline will converge to a certain extent. It is difficult for SLB to reduce the freight greatly, which has a higher unit cost and a smaller profit margin. So, when the relative freight is too low because of the increase in the SLB's substitution effect, the much better alternative for SLB is to relinquish the joint quotation strategy. At the same time, the significant reduction in the relative freight is also unfavorable to SLA. Due to the container shipping industry characteristics of freight dominance and long-term contracted demand, the low freight of SLB has a negative impact on the normal quotation of SLA. Therefore, under the joint quotation strategy, the increases in service substitution effect from SLB will be adverse for the shipping blockchain alliance. The increases in service substitution effect from SLA will be good for the shipping blockchain alliance. However, compared with SLA, the change in competition intensity of SLB has a more substantial impact on the stability of a shipping blockchain alliance.

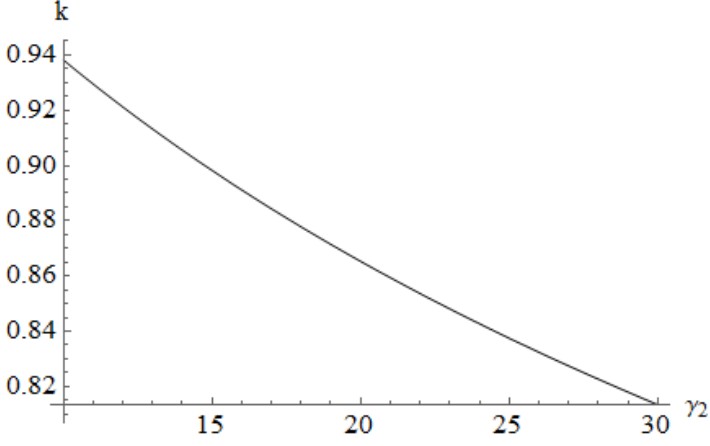

**Figure 6.** The relative freight sensitivity of $\gamma_2$.

### 4.2.2. The Profit Sensitivities on the Competition Intensity Coefficients

According to Figure 7a,b, both SLA's and SLB's profits at the initial state are negatively affected by the improving of SLB's service substitution effect. While SLA's profit under the joint quotation strategy ($\pi_1^{JQ}$) is more stable and is higher than the initial state when $\gamma_1$ is

reasonable and appropriately increased, SLA's profit in the initial state seems to be higher when $\gamma_1$ too small. SLB's profit under the joint quotation strategy ($\pi_2^{JQ}$) is more sensitive to the change of $\gamma_1$. However, the joint quotation strategy will usually ensure that SLB obtains an excess profit better than the Initial Theoretical Equilibrium state when $\gamma_1$ is not too high.

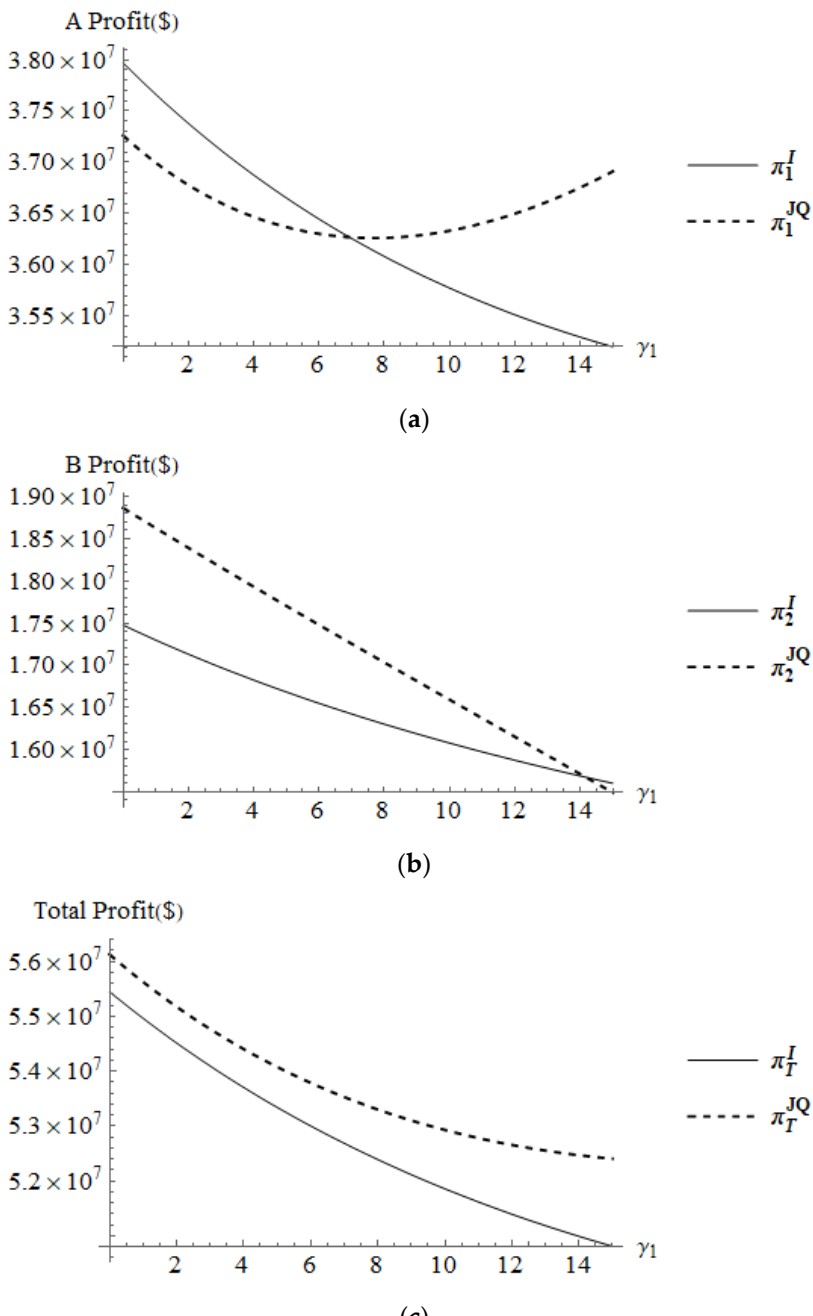

**Figure 7.** The profit sensitivities of $\gamma_1$. (**a**) SLA's profit sensitivities of $\gamma_1$. (**b**) SLB's profit sensitivities of $\gamma_1$. (**c**) Total profit sensitivities of $\gamma_1$.

According to Figure 7c, the total profit under the joint quotation strategy ($\pi_T^{JQ}$) is always higher than that under the Initial Theoretical Equilibrium state ($\pi_T^I$). Additionally, it can be seen intuitively from the trends of the two curves that the improvement of the service substitution of SLB is favorable to stabilize the total profit.

The improvement of SLB's service substitution effect is more beneficial for SLA to stabilize profit and achieve excess profit under the joint quotation strategy. Additionally, it

is also favorable for the shipping blockchain alliance to obtain a more stable and higher-level total profit. Although the increase in $\gamma_1$ makes SLB's own profit under the joint quotation strategy more unstable, it can at least help SLB to obtain a more substantial excess profit than the Initial Theoretical Equilibrium state. Therefore, the improvement in the SLB's service substitution effect under the joint quotation strategy is beneficial for the sustainable development of the shipping blockchain alliance.

According to Figure 8a, the improvement in the SLA's service substitution effect is unfavorable for its own profit no matter whether it is under the joint quotation strategy ($\pi_1^{JQ}$) or in the Initial Theoretical Equilibrium state ($\pi_1^I$). Moreover, SLA's profit is more sensitive to the change in $\gamma_2$ under the joint quotation strategy than in the Initial Theoretical Equilibrium state. The enhancement of SLA's service substitution effect is detrimental to its own stable profit under the joint quotation strategy. When $\gamma_2$ reaches a certain level, SLA will not continue to obtain excess profit and even face a worse profit level than the initial state if it continues to adopt the joint quotation strategy.

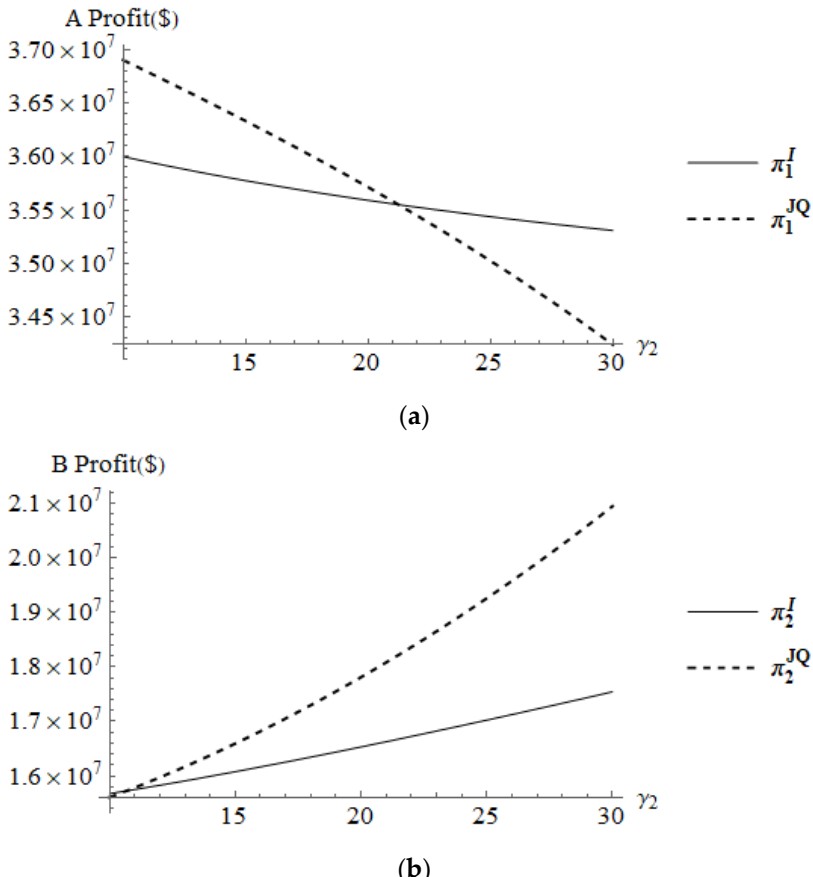

**Figure 8.** *Cont.*

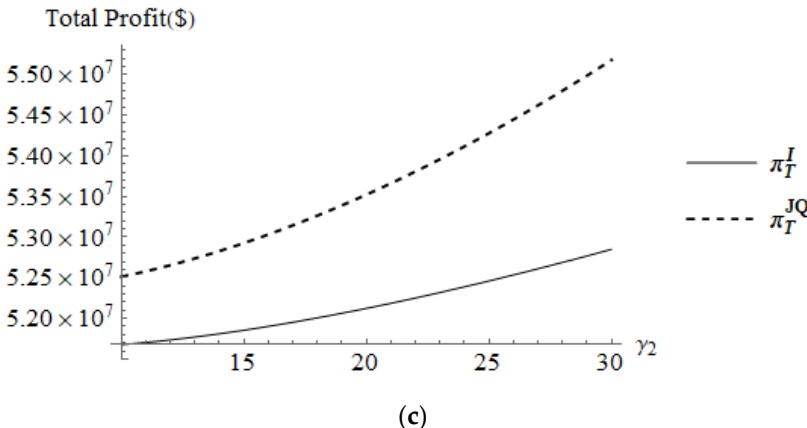

**(c)**

**Figure 8.** The profit sensitivities of $\gamma_2$. (**a**) SLA's profit sensitivities of $\gamma_2$. (**b**) SLB's profit sensitivities of $\gamma_2$. (**c**) Total profit sensitivities of $\gamma_2$.

In accordance with Figure 8b, SLB's profit under the joint quotation strategy ($\pi_2^{JQ}$) is more sensitive to the SLA's service substitution effect than that in the Initial Theoretical Equilibrium state ($\pi_2^I$). The profit level of SLB under the joint quotation strategy is significantly higher than the initial state. The total profit under the joint quotation strategy ($\pi_T^{JQ}$) is also more sensitive to SLA's service substitution effect than that in the Initial Theoretical Equilibrium state ($\pi_T^I$), on the basis of Figure 8c. Additionally, the total profit under the Joint Quotation strategy is higher than the initial state.

The improvement in the SLA's service substitution effect is beneficial for SLB and the shipping blockchain alliance achieving excess profits. However, it is quite unfavorable for SLA itself. However, the change in the SLA' s service substitution effect will bring significantly negative effects for the profits of SLA and SLB and the total profit, because it will increase the profit instability of the SLs and the shipping blockchain alliance under the joint quotation strategy. Therefore, the change in the SLA's service substitution effect is unfavorable to SLA or SLB or the shipping blockchain alliance from the perspective of stable profit.

### 4.2.3. The Demand Sensitivities on the Competition Intensity Coefficients

According to Figure 9a, the demand of SLA under the joint quotation strategy ($D_1^{JQ}$) and in the Initial Theoretical Equilibrium state ($D_1^I$) are both positively affected by $\gamma_1$. However, the demand level under the joint quotation strategy is much lower than that in the Initial Theoretical Equilibrium state.

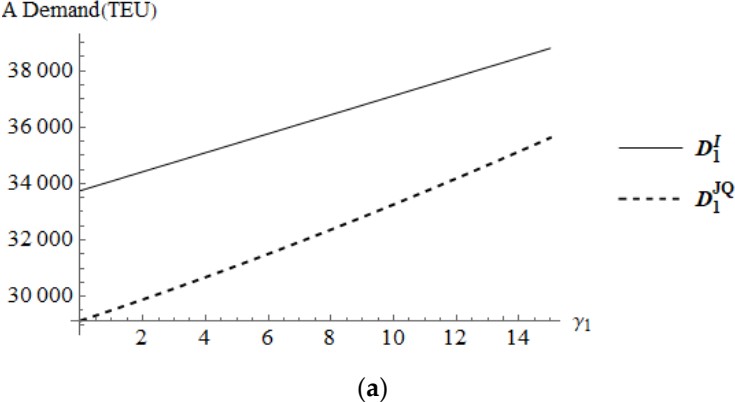

**(a)**

**Figure 9.** *Cont.*

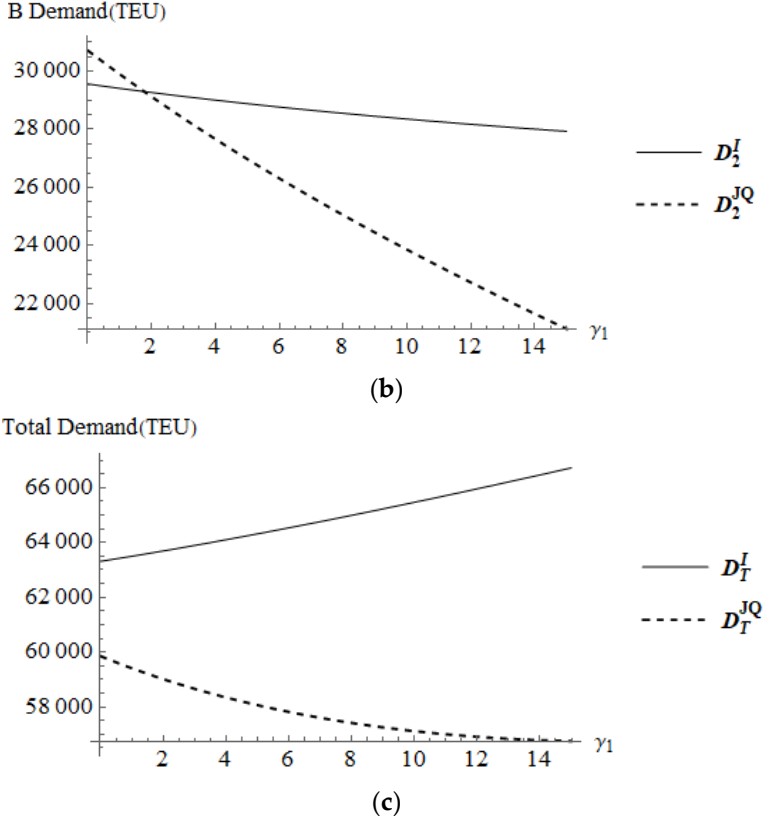

**Figure 9.** The demand sensitivities of $\gamma_1$. (**a**) SLA's demand sensitivities of $\gamma_1$. (**b**) SLB's demand sensitivities of $\gamma_1$. (**c**) Total demand sensitivities of $\gamma_1$.

According to Figure 9b, SLB's demand in the Initial Theoretical Equilibrium state ($D_2^I$) is little negatively affected by $\gamma_1$. SLB's demand under the joint quotation strategy ($D_2^{JQ}$) is negatively affected by $\gamma_1$. Additionally, SLB's demand level under the joint quotation strategy will be lower than that under the Initial Theoretical Equilibrium state at the time $\gamma_1 > 2$. That means that the demand of SLB will be greatly reduced with the increase in its own competition under the joint quotation strategy.

According to Figure 9c, the total demand under the Initial Theoretical Equilibrium state ($D_T^I$) is positively affected by $\gamma_1$, but the total demand under the joint quotation strategy ($D_T^{JQ}$) is negatively affected by $\gamma_1$. However, the total demand level under the Initial Theoretical Equilibrium state is always much higher than that under the joint quotation strategy.

According to Figure 10a, SLA's demand under the Initial Theoretical Equilibrium state ($D_1^I$) is negatively affected slightly by $\gamma_2$, while SLA's demand under the joint quotation strategy ($D_1^{JQ}$) is negatively affected significantly by $\gamma_2$. SLA's demand level under the Initial Theoretical Equilibrium state is always higher than that under the joint quotation strategy. However, it is obvious that SLA's demand under the joint quotation strategy is more sensitive to the change in its own service substitution effect.

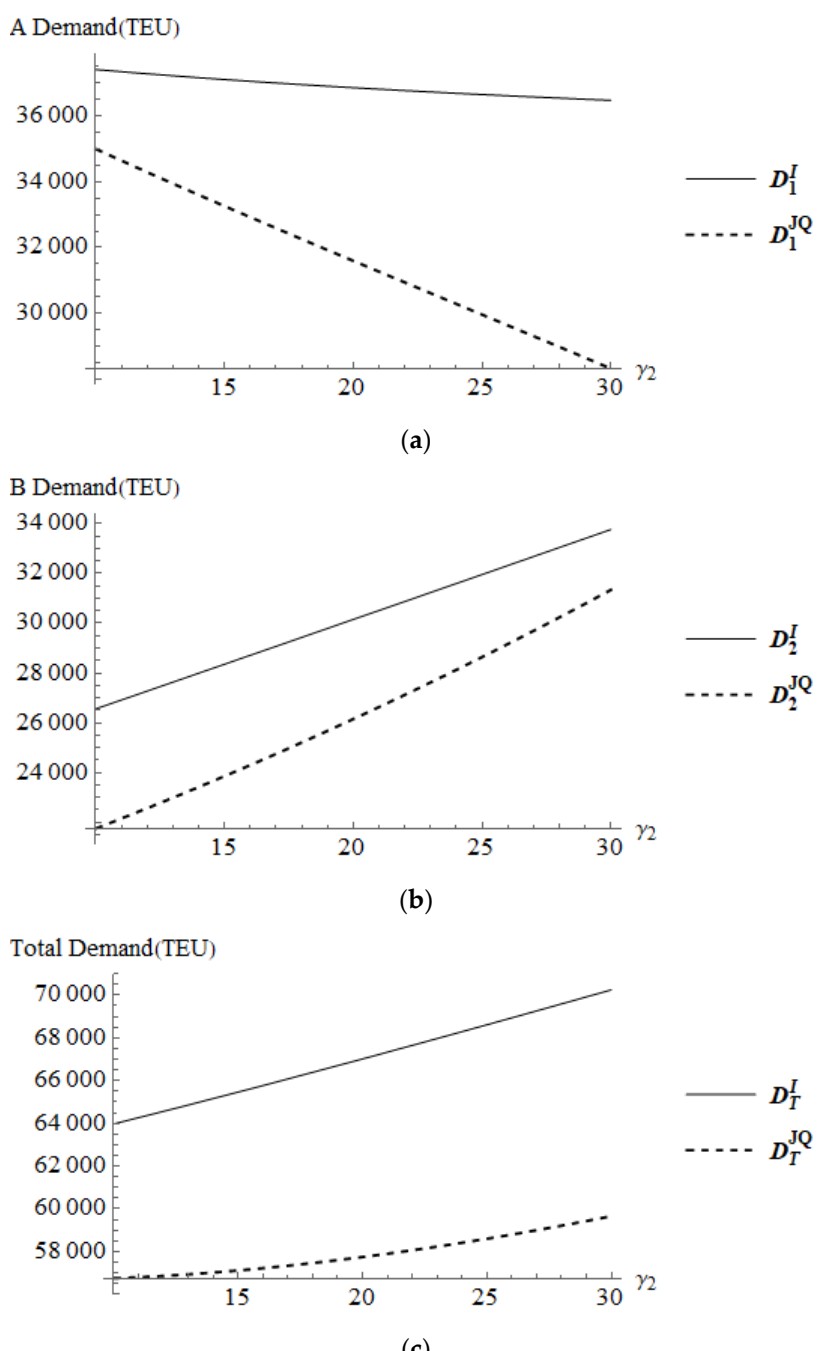

**Figure 10.** The demand sensitivities of $\gamma_2$. (**a**) SLA's demand sensitivities of $\gamma_2$. (**b**) SLB's demand sensitivities of $\gamma_2$. (**c**) Total demand sensitivities of $\gamma_2$.

According to Figure 10b, the demands of SLB under the joint quotation ($D_2^{JQ}$) and the Initial Theoretical Equilibrium state ($D_2^{I}$) are both positively affected by $\gamma_2$. The demand level of SLB under the Initial Theoretical Equilibrium state is higher than that under the joint quotation strategy. Additionally, the demand of shipping line B under the joint quotation strategy is more sensitive to the change of SLA's service substitution effect.

According to Figure 10c, total demands under the joint quotation strategy ($D_T^{JQ}$) and the Initial Theoretical Equilibrium state ($D_T^{I}$) are both positively affected by $\gamma_2$. However, total demand in the Initial Theoretical Equilibrium state is more sensitive to the change in SLA's service substitution effect than that under the joint quotation strategy. Additionally, the total demand level in the Initial Theoretical Equilibrium state is always higher than

that under the joint quotation strategy. Under the joint quotation strategy, the effects of the same competition intensity coefficient on two SLs' demands are quite opposite. The demand of one SL is usually negatively affected by the change of its own competitiveness but positively affected by the change in the competitor's competitiveness. Although the larger SL's competitiveness can positively impact the total demand, it is less sensitive than in the Initial Theoretical Equilibrium state. In other words, the joint quotation strategy can help to reduce the fluctuation of overall total demand caused by changes in internal competition. However, the cost is a relatively lower-level total demand compared with the Initial Theoretical Equilibrium state.

## 5. Conclusions

At present, more and more factors affect the development of the international container shipping industry, such as geopolitics and COVID-19. Affected by the long-term volatility of international trade, the container shipping industry has shifted from excess transportation capacity to insufficient capacity in recent years. Both situations are very detrimental to the sustainable development of the container shipping industry. In order to cope with the low operational efficiency of the shipping industry caused by the changes in many external factors, more and more shipping lines try to build a shipping blockchain alliance to improve their risk response ability. The public quotation mechanism based on blockchain has also become a possibility for the future development of the shipping blockchain because it can help shipping lines to standardize their quotations. Under the context, the feasibility of further establishing the joint quotation strategy on the shipping blockchain to help shipping lines to stabilize demand and achieve excess profits was discussed in this paper. Aiming to solve the problem in this paper, the equivalent freight method was introduced. Additionally, a multi-round fixed proportion joint quotation model was constructed. At the same time, numerical experiments and sensitivity analysis were introduced to explore the impact of internal competition changes in the shipping blockchain alliance on the optimal strategy and stability of the joint quotation.

There are some interesting findings in this paper. From the perspective of realizing excess profit, the joint quotation strategy can indeed ensure that each shipping line obtain the excess profit and maximize the total profit of the alliance at the same time. For shipping lines in the same shipping blockchain alliance, the intensification of internal competition is conducive to stabilizing the profit level under the optimal freight rate strategy of the joint quotation to a certain extent. It reflects the stability of the joint quotation equilibrium strategy and the sustainability of improving the shipping lines' profit. However, the stability is conditional. In other words, the performance of the joint quotation strategy for stabilizing the profit of decision-makers can be reflected when the service substitution effect of competitors in the same alliance is enhanced. On the contrary, when the decision-maker's own service substitution effect is enhanced, its profit may deteriorate under the joint quotation strategy. The deterioration is reflected in two aspects, (1) the volatility of profit increases and (2) the profit level decreases significantly. Additionally, it may even be lower than the Initial Theoretical Equilibrium state.

Combined with numerical experiments, it is not difficult to see that the joint quotation strategy improves the profit by increasing the equilibrium freight rate level. However, for the container shipping industry, which is dominated by long-term contracted demand, it may mean a substantial decline in the level of demand. In addition, under the blockchain public quotation mechanism, the adjustment of relative freight rate may also cause substantial fluctuations in container shipping demand. Theoretically, the equilibrium achieved by greatly increasing the freights may be unstable. On one hand, the joint quotation based on the shipping blockchain actually forms a monopoly effect on the freight rate. Compared with the traditional shipping market, the strong bargaining power of container shipping lines and the excessively elevated freight rate level may deter shippers. Additionally, the joint quotation may also be resolutely resisted by the shipper due to monopoly risk, resulting in a major risk of sharp reduction in demand. So, the correct development direction



of the shipping blockchain in future is to build a fair competition relationship among members. It should not only put an end to vicious price competition, but also resolutely prevent the risk of collusion and monopoly of shipping giants under the information disclosure mechanism, because the monopoly is not conducive to the long-term sustainable development of the shipping blockchain, either in terms of law or in terms of the benefits studied in this paper.

The findings in this paper can help container shipping alliances registered with blockchains to optimize their freight strategies on the emerging blockchain market. However, there are some limitations in this paper. The number of shipping blockchain alliance members in the case is only two. Additionally, the decision-making system is simple. Only the multi-round dynamic decision-making process under the premise of cooperation is considered. In addition, the demand function applied can be further enriched by considering the seasonality and randomness characteristics. In the future, it can be further investigated that whether the members of the shipping blockchain alliance can deepen cooperation and strengthen risk response ability through various ways in the dynamic decision-making process. Additionally, the properties of the demand function can be extended from linear and binary to nonlinear and multivariate.

**Author Contributions:** Conceptualization, F.Z. and Y.G.; methodology and software, F.Z.; validation, F.Z. and Y.G.; writing—original draft preparation, F.Z.; writing—review and editing, F.Z. and Y.G.; supervision, Y.G. All authors have read and agreed to the published version of the manuscript.

**Funding:** The manuscript received no external funding.

**Institutional Review Board Statement:** Not applicable.

**Informed Consent Statement:** Not applicable.

**Data Availability Statement:** Not applicable.

**Conflicts of Interest:** The authors declare no conflict of interest.

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
