# Peer review of "Approach to an Equivalent Freight-Based Sustainable Joint-Quotation Strategy for Shipping Blockchain Alliance"

_sustainability, doi:10.3390/su141610441_

Round 1

Reviewer 1 Report

The paper deals with a very actual and interesting topic. It seems that all analysis is scientifically valid and technically accurate. The findings are also reproducible.

However, in the intro sections, there are a number of strange statements that need checking / updating. Besides, the authors need to address the following comments to improve the presentation quality of the paper. Firstly, the authors need to supplement the operation mechanism of the shipping blockchain freight open market to highlight the practical significance of the issues studied in this paper. Secondly, the authors need to add more joint quotation related work in section 2 in order to comprehensively demonstrate the research status of the joint quotation, and identify the research gap. Thirdly, the authors need to include a table covering the units of the corresponding metrics in the paper. Besides, the authors need to make a clear explanation on why the container shipping industry is focused in the paper, dose the problem also exist in the field of bulk shipping? The overall paper needs to be checked for typos, syntax, and grammar errors in order to improve the quality of its presentation. My indication is minor revisions.

Author Response

Reviewer 1’s comments and suggestions for the authors as follows:

The paper deals with a very actual and interesting topic. It seems that all analysis is scientifically valid and technically accurate. The findings are also reproducible.

However, in the intro sections, there are a number of strange statements that need checking / updating. Besides, the authors need to address the following comments to improve the presentation quality of the paper.

  • Firstly, the authors need to supplement the operation mechanism of the shipping blockchain freight open market to highlight the practical significance of the issues studied in this paper.

Our Responses: Thank you very much for your professional advices. According to your valuable comments, the corresponding complement has been made in section 3 Model Development involved in yellow highlight.

  • Secondly, the authors need to add more joint quotation related work in section 2 in order to comprehensively demonstrate the research status of the joint quotation, and identify the research gap.

Our Responses: Thank you very much for your professional advices. According to your valuable comments, the corresponding complement has been made in section 2 Literature Review involved in yellow highlight.

  • Thirdly, the authors need to include a table covering the units of the corresponding metrics in the paper.

Response: Thank you very much for your professional advices. According to your valuable comments, the corresponding complement has been made in section 3.1 Basic Assumptions Table 1 involved in yellow highlight.

  • Besides, the authors need to make a clear explanation on why the container shipping industry is focused in the paper, dose the problem also exist in the field of bulk shipping?

Our Responses: Thank you very much for your professional advices. According to your valuable comments, the corresponding explanation has been made in section 3 Model Development Involved in yellow highlight.

  • The overall paper needs to be checked for typos, syntax, and grammar errors in order to improve the quality of its presentation. My indication is minor revisions.

Our Responses: Thank you very much for your professional advices. According to your valuable comments, we have checked and made modification on typos, syntax, and grammar errors of the paper.

Reviewer 2 Report

The authors focus their study on the freight decision-making problem of container shipping lines in the same shipping blockchain alliance. Towards this direction, the authors introduce a multi-round joint quotation framework with the concept of relative freight and fixed proportion pricing method based on the emerging shipping blockchain market. The research problem, method and the model in this paper reflect a certain degree of innovation. The authors have successfully verified the positive role of the joint quotation strategy for shipping blockchain members to obtain excess profits and stabilize demand. The conclusion of the manuscript is in line with general cognition, and also reflects a strong practical significance. In addition, the manuscript is overall well written and easy to follow and the authors have well thought out their main contributions. The provided theoretical analysis is concrete, complete, and correct and the authors have provided all the intermediate steps in order to enable the average reader to easily follow it. The provided numerical results are also rich in order to show the pure operation and the performance of the proposed framework.

However, the authors are encouraged to consider the following suggestions provided by the reviewer in order to improve the standardization and readability of their manuscript.

(1) In the literature review part, it will be better for the authors to take use of more summative language and identify the research gap that the authors try to address.

(2) In the sensitivity analysis part, it will be better for the authors to adjust the title of the figure to the expression, for example, adjust the form Figure 8a. SLA’s Profit Sensitivities… to the form a). SLA’s Profit Sensitivities….

(3) There are many inappropriate expressions in the manuscript. It will be better for the authors to carefully check the expression and typos of the manuscript to improve accuracy and readability,

Author Response

Reviewer 2’s comments and suggestions for the authors as follows:

The authors focus their study on the freight decision-making problem of container shipping lines in the same shipping blockchain alliance. Towards this direction, the authors introduce a multi-round joint quotation framework with the concept of relative freight and fixed proportion pricing method based on the emerging shipping blockchain market. The research problem, method and the model in this paper reflect a certain degree of innovation. The authors have successfully verified the positive role of the joint quotation strategy for shipping blockchain members to obtain excess profits and stabilize demand. The conclusion of the manuscript is in line with general cognition, and also reflects a strong practical significance. In addition, the manuscript is overall well written and easy to follow and the authors have well thought out their main contributions. The provided theoretical analysis is concrete, complete, and correct and the authors have provided all the intermediate steps in order to enable the average reader to easily follow it. The provided numerical results are also rich in order to show the pure operation and the performance of the proposed framework.

However, the authors are encouraged to consider the following suggestions provided by the reviewer in order to improve the standardization and readability of their manuscript.

(1) In the literature review part, it will be better for the authors to take use of more summative language and identify the research gap that the authors try to address.

Our Responses: Thank you very much for your professional advices. According to your valuable comments, the complement has been made in section 2 involved in yellow highlight.

(2) In the sensitivity analysis part, it will be better for the authors to adjust the title of the figure to the expression, for example, adjust the form Figure 8a. SLA’s Profit Sensitivities… to the form a). SLA’s Profit Sensitivities…

Our Responses: Thank you very much for your professional advices. According to your valuable comments, we have made the revision and modification in Section 4.2 involved in yellow highlight.

(3) There are many inappropriate expressions in the manuscript. It will be better for the authors to carefully check the expression and typos of the manuscript to improve accuracy and readability,

Our Responses: Thank you very much for your professional advices. According to your valuable comments, we have checked and made modification on typos, syntax, and grammar errors of the paper.
